# Better Private Linear Regression Through Better Private Feature Selection

Travis Dick[*]        Jennifer Gillenwater[†]        Matthew Joseph[‡]

## Abstract

Existing work on differentially private linear regression typically assumes that end users can precisely set data bounds or algorithmic hyperparameters. End users often struggle to meet these requirements without directly examining the data (and violating privacy). Recent work has attempted to develop solutions that shift these burdens from users to algorithms, but they struggle to provide utility as the feature dimension grows. This work extends these algorithms to higher-dimensional problems by introducing a differentially private feature selection method based on Kendall rank correlation. We prove a utility guarantee for the setting where features are normally distributed and conduct experiments across 25 datasets. We find that adding this private feature selection step before regression significantly broadens the applicability of "plug-and-play" private linear regression algorithms at little additional cost to privacy, computation, or decision-making by the end user.

## 1   Introduction

Differentially private [10] algorithms employ carefully calibrated randomness to obscure the effect of any single data point. Doing so typically requires an end user to provide bounds on input data to ensure the correct scale of noise. However, end users often struggle to provide such bounds without looking at the data itself [27], thus nullifying the intended privacy guarantee. This has motivated the development of differentially private algorithms that do not require these choices from users.

To the best of our knowledge, two existing differentially private linear regression algorithms satisfy this "plug-and-play" requirement: 1) the Tukey mechanism [2], which combines propose-test-release with an exponential mechanism based on Tukey depth, and 2) Boosted AdaSSP [30], which applies gradient boosting to the AdaSSP algorithm introduced by Wang [34]. We refer to these methods as, respectively, Tukey and BAS. Neither algorithm requires data bounds, and both feature essentially one chosen parameter (the number of models $m$ for Tukey, and the number of boosting rounds $T$ for BAS) which admits simple heuristics without tuning. In contrast, AdaSSP requires a user to provide bounds on the data's feature and label norms, and DPSGD requires a user to configure hyperparameters including learning rate, clipping norms, batch size, and number of epochs. Both algorithms produce much weaker utility when these parameters are even moderately misconfigured [2].

Tukey obtains strong empirical results when the number of data points $n$ greatly exceeds the feature dimension $d$ [2], while BAS obtains somewhat weaker utility on a larger class of datasets [30]. Nonetheless, neither algorithm provides generally strong performance on its own. Evaluated over a collection of 25 linear regression datasets taken from Tang et al. [30], Tukey and BAS obtain coefficient of determination $R^2 > 0$ on only four (see Section 4); for context, a baseline of $R^2 = 0$ is achieved by the trivial constant predictor, which simply outputs the mean label. These results suggest room for improvement for practical private linear regression.

---

[*]tdick@google.com, Google Research

[†]jgillenw@gmail.com, work done while at Google Research

[‡]mtjoseph@google.com, Google Research

37th Conference on Neural Information Processing Systems (NeurIPS 2023).

## 1.1 Our Contributions

We extend existing work on private linear regression by adding a preprocessing step that applies private feature selection. At a high level, this strategy circumvents the challenges of large feature dimension $d$ by restricting attention to $k \ll d$ carefully selected features. We initiate the study of private feature selection in the context of "plug-and-play" private linear regression and make two concrete contributions:

1. We introduce a practical algorithm, DPKendall, for differentially private feature selection (Section 3). DPKendall uses Kendall rank correlation [20] and only requires the user to choose the number $k$ of features to select. It satisfies $\varepsilon$-DP and, given $n$ samples with $d$-dimensional features, runs in time $O(dkn\log(n))$ (Theorem 3.4). We also provide a utility guarantee when the features are normally distributed (Theorem 3.8).

2. We conduct experiments across 25 datasets (Section 4), with $k$ fixed at 5 and 10. These compare Tukey and BAS without feature selection, with SubLasso feature selection [21], and with DPKendall feature selection. Using $(\ln(3), 10^{-5})$-DP to cover both private feature selection and private regression, we find at $k = 5$ that adding DPKendall yields $R^2 > 0$ on 56% of the datasets. Replacing DPKendall with SubLasso drops the rate to 40% of datasets, and omitting feature selection entirely drops it further to 16%.

In summary, we suggest that DPKendall significantly expands the applicability and utility of practical private linear regression.

## 1.2 Related Work

The focus of this work is practical private feature selection applied to private linear regression. We therefore refer readers interested in a more general overview of the private linear regression literature to the discussions of Amin et al. [2] and Tang et al. [30].

Several works have studied private sparse linear regression [21, 31, 17, 29]. However, Jain and Thakurta [17] and Talwar et al. [29] require an $\ell_\infty$ bound on the input data, and the stability test that powers the feature selection algorithm of Thakurta and Smith [31] requires the end user to provide granular details about the optimal Lasso model. These requirements are significant practical obstacles. An exception is the work of Kifer et al. [21]. Their algorithm first performs feature selection using subsampling and aggregation of non-private Lasso models. This feature selection method, which we call SubLasso, only requires the end user to select the number of features $k$. To the selected features, Kifer et al. [21] then apply objective perturbation to privately optimize the Lasso objective. As objective perturbation requires the end user to choose parameter ranges and provides a somewhat brittle privacy guarantee contingent on the convergence of the optimization, we do not consider it here. Instead, our experiments combine SubLasso feature selection with the Tukey and BAS private regression algorithms. An expanded description of SubLasso appears in Section 4.2.

We now turn to the general problem of private feature selection. There is a significant literature studying private analogues of the general technique of principal component analysis (PCA) [25, 15, 8, 19, 11, 1]. Unfortunately, all of these algorithms assume some variant of a bound on the row norm of the input data. Stoddard et al. [28] studied private feature selection in the setting where features and labels are binary, but it is not clear how to extend their methods to the non-binary setting that we consider in this work. SubLasso is therefore the primary comparison private feature selection method in this paper. We are not aware of existing work that studies private feature selection in the specific context of "plug-and-play" private linear regression.

Finally, private rank correlation has previously been studied by Kusner et al. [23]. They derived a different, normalized sensitivity bound appropriate for their "swap" privacy setting and applied it to privately determine the causal relationship between two random variables.

## 2 Preliminaries

Throughout this paper, a database $D$ is a collection of labelled points $(x, y)$ where $x \in \mathbb{R}^d$, $y \in \mathbb{R}$, and each user contributes a single point. We use the "add-remove" form of differential privacy.

**Definition 2.1** ([10]). *Databases $D, D'$ from data domain $\mathcal{D}$ are* neighbors $D \sim D'$ *if they differ in the presence or absence of a single record. A randomized mechanism $\mathcal{M} : \mathcal{D} \to \mathcal{O}$ is $(\varepsilon, \delta)$-differentially private (DP) if for all $D \sim D' \in \mathcal{D}$ and any $S \subseteq \mathcal{O}$*

$$\mathbb{P}_{\mathcal{M}} [\mathcal{M}(D) \in S] \le e^{\varepsilon} \mathbb{P}_{\mathcal{M}} [\mathcal{M}(D') \in S] + \delta.$$

We use basic composition to reason about the privacy guarantee obtained from repeated application of a private algorithm. More sophisticated notions of composition exist, but for our setting of relatively few compositions, basic composition is simpler and suffers negligible utility loss.

**Lemma 2.2** ([10]). *Suppose that for $j \in [k]$, algorithm $\mathcal{A}_j$ is $(\varepsilon_j, \delta_j)$-DP. Then running all $k$ algorithms is $(\sum_{j=1}^{k} \varepsilon_j, \sum_{j=1}^{k} \delta_j)$-DP.*

Both SubLasso and DPKendall use a private subroutine for identifying the highest count item(s) from a collection, known as private top-$k$. Several algorithms for this problem exist [4, 9, 13]. We use the pure DP "peeling mechanism" based on Gumbel noise [9], as its analysis is relatively simple, and its performance is essentially identical to other variants for the relatively small $k$ used in this paper.

**Definition 2.3** ([9]). *A Gumbel distribution with parameter $b$ is defined over $x \in \mathbb{R}$ by $\mathbb{P}[x; b] = \frac{1}{b} \cdot \exp\left(-\frac{x}{b} - e^{-x/b}\right)$. Given $c = (c_1, \ldots, c_d) \in \mathbb{R}^d$, $k \in \mathbb{N}$, and privacy parameter $\varepsilon$, $\mathsf{Peel}(c, k, \Delta_{\infty}, \varepsilon)$ adds independent $\mathsf{Gumbel}\left(\frac{2k\Delta_{\infty}}{\varepsilon}\right)$ noise to each count $c_j$ and outputs the ordered sequence of indices with the largest noisy counts.*

**Lemma 2.4** ([9]). *Given $c = (c_1, \ldots, c_d) \in \mathbb{R}^d$ with $\ell_{\infty}$ sensitivity $\Delta_{\infty}$, $\mathsf{Peel}(c, k, \Delta_{\infty}, \varepsilon)$ is $\varepsilon$-DP.*

The primary advantage of Peel over generic noise addition is that, although users may contribute to $d$ counts, the added noise only scales with $k$. We note that while Peel requires an $\ell_{\infty}$ bound, neither DPKendall nor SubLasso needs user input to set it: regardless of the dataset, DPKendall's use of Peel has $\ell_{\infty} = 3$ and SubLasso's use has $\ell_{\infty} = 1$ (see Algorithms 1 and 2).

# 3 Feature Selection Algorithm

This section describes our feature selection algorithm, DPKendall, and formally analyzes its utility. Section 3.1 introduces and discusses Kendall rank correlation, Section 3.2 describes the full algorithm, and the utility result appears in Section 3.3.

## 3.1 Kendall Rank Correlation

The core statistic behind our algorithm is Kendall rank correlation. Informally, Kendall rank correlation measures the strength of a monotonic relationship between two variables.

**Definition 3.1** ([20]). *Given a collection of data points $(X, Y) = \{(X_1, Y_1), \ldots, (X_n, Y_n)\}$ and $i < i'$, a pair of observations $(X_i, Y_i), (X_{i'}, Y_{i'})$ is* discordant *if $(X_i - X_{i'})(Y_i - Y_{i'}) < 0$. Given data $(X, Y)$, let $d_{X,Y}$ denote the number of discordant pairs. Then the* empirical Kendall rank correlation *is*

$$\hat{\tau}(X, Y) := \frac{n}{2} - \frac{2 d_{X,Y}}{n - 1}.$$

*For real random variables $X$ and $Y$, we can also define the* population Kendall rank correlation *by*

$$\tau(X, Y) = \mathbb{P}\left[(X - X')(Y - Y') > 0\right] - \mathbb{P}\left[(X - X')(Y - Y') < 0\right].$$

Kendall rank correlation is therefore high when an increase in $X$ or $Y$ typically accompanies an increase in the other, low when an increase in one typically accompanies a decrease in the other, and close to 0 when a change in one implies little about the other. We typically focus on empirical Kendall rank correlation, but the population definition will be useful in the proof of our utility result.

Before discussing Kendall rank correlation in the context of privacy, we note two straightforward properties. First, for simplicity, we use a version of Kendall rank correlation that does not account for ties. We ensure this in practice by perturbing data by a small amount of continuous random noise. Second, $\tau$ has range $[-1, 1]$, but (this paper's version of) $\hat{\tau}$ has range $[-n/2, n/2]$. This scaling does not affect the qualitative interpretation and ensures that $\hat{\tau}$ has low sensitivity[4].

---

[4] In particular, without scaling, $\hat{\tau}$ would be $\approx \frac{1}{n}$-sensitive, but $n$ is private information in add-remove privacy.

**Lemma 3.2.** $\Delta(\hat{\tau}) = 3/2$.

*Proof.* At a high level, the proof verifies that the addition or removal of a user changes the first term of $\hat{\tau}$ by at most 1/2, and the second term by at most 1.

In more detail, consider two neighboring databases $(X, Y)$ and $(X', Y')$. Without loss of generality, we may assume that $(X, Y) = \{(X_1, Y_1), \ldots, (X_n, Y_n)\}$ and $(X', Y') = \{(X'_1, Y'_1), \ldots, (X'_{n+1}, Y'_{n+1})\}$ where for all $i \in [n]$ we have that $X'_i = X_i$ and $Y'_i = Y_i$. First, we argue that the number of discordant pairs in $(X', Y')$ cannot be much larger than in $(X, Y)$. By definition, we have that $d_{X',Y'} - d_{X,Y} = \sum_{j=1}^{n} \mathbb{1}_{(X_j - X'_{n+1})(Y_j - Y'_{n+1}) < 0}$. In particular, this implies that $d_{X',Y'} - d_{X,Y} \in [0, n]$.

We can rewrite the difference in Kendall correlation between $(X, Y)$ and $(X', Y')$ as follows:

$$
\begin{aligned}
\hat{\tau}(X, Y) - \hat{\tau}(X', Y') &= \frac{n}{2} - \frac{2d_{X,Y}}{n-1} - \frac{n+1}{2} + \frac{2d_{X',Y'}}{n} \\
&= 2\left(\frac{d_{X',Y'}}{n} - \frac{d_{X,Y}}{n-1}\right) - \frac{1}{2} \\
&= 2\left(\frac{d_{X',Y'} - d_{X,Y}}{n} + \frac{d_{X,Y}}{n} - \frac{d_{X,Y}}{n-1}\right) - \frac{1}{2} \\
&= 2\frac{d_{X',Y'} - d_{X,Y}}{n} + 2d_{X,Y}\left(\frac{1}{n} - \frac{1}{n-1}\right) - \frac{1}{2} \\
&= 2\frac{d_{X',Y'} - d_{X,Y}}{n} - \frac{d_{X,Y}}{\binom{n}{2}} - \frac{1}{2},
\end{aligned}
$$

where the final equality follows from the fact that $2(\frac{1}{n} - \frac{1}{n-1}) = -1/\binom{n}{2}$. Using our previous calculation, the first term is in the range $[0, 2]$ and, since $d_{X,Y} \in [0, \binom{n}{2}]$, the second term is in the range $[-1, 0]$. It follows that

$$
-\frac{3}{2} = 0 - 1 - \frac{1}{2} \le \hat{\tau}(X, Y) - \hat{\tau}(X', Y') \le 2 - 0 - \frac{1}{2} = \frac{3}{2},
$$

and therefore $|\hat{\tau}(X, Y) - \hat{\tau}(X', Y')| \le 3/2$.

To show that the sensitivity is not smaller than $3/2$, consider neighboring databases $(X, Y)$ and $(X', Y')$ such that $d_{X,Y} = 0$ and $(X', Y')$ contains a new point that is discordant with all points in $(X, Y)$. Then $d_{X',Y'} = n$ while $d_{X,Y} = 0$. Then $\hat{\tau}(X, Y) - \hat{\tau}(X', Y') = 2 - 0 - 1/2 = 3/2$. □

Turning to privacy, Kendall rank correlation has two notable strengths. First, because it is computed entirely from information about the relative ordering of data, it does not require an end user to provide data bounds. This makes it a natural complement to private regression methods that also operate without user-provided data bounds. Second, Kendall rank correlation's sensitivity is constant, but its range scales linearly with $n$. This makes it easy to compute privately. A contrasting example is Pearson correlation, which requires data bounds to compute covariances and has sensitivity identical to its range. An extended discussion of alternative notions of correlation appears in Section 6.

Finally, Kendall rank correlation can be computed relatively quickly using a variant of merge sort.

**Lemma 3.3** ([22]). *Given collection of data points $(X, Y) = \{(X_1, Y_1), \ldots, (X_n, Y_n)\}$, $\hat{\tau}(X, Y)$ can be computed in time $O(n \log(n))$.*

### 3.2 DPKendall

Having defined Kendall rank correlation, we now describe our private feature selection algorithm, DPKendall. Informally, DPKendall balances two desiderata: 1) selecting features that correlate with the label, and 2) selecting features that do not correlate with previously selected features. Prioritizing only the former selects for redundant copies of a single informative feature, while prioritizing only the latter selects for features that are pure noise.

In more detail, DPKendall consists of $k$ applications of Peel to select a feature that is correlated with the label and relatively uncorrelated with the features already chosen. Thus, letting $S_t$ denote the set

of $t-1$ features already chosen in round $t$, each round attempts to compute

$$\max_{j \notin S_t} \left( |\hat{\tau}(X_j, Y)| - \frac{1}{t-1} \sum_{j' \in S_t} |\hat{\tau}(X_j, X_{j'})| \right). \tag{1}$$

The $\frac{1}{t-1}$ scaling ensures that the sensitivity of the overall quantity remains fixed at $\frac{3}{2}$ in the first round and 3 in the remaining rounds. Note that in the first round we take second term to be 0, and only label correlation is considered.

---

**Algorithm 1** DPKendall$(D, k, \varepsilon)$

---

1: **Input:** Examples $D = \{(X_i, Y_i)\}_{i=1}^n$, number of selected features $k$, privacy parameter $\varepsilon$
2: **for** $j = 1, \ldots, d$ **do**
3:      Compute $\hat{\tau}_j^Y = |\hat{\tau}(X_j, Y)|$
4: Initialize $S = \emptyset$
5: Initialize $\hat{\tau} = \hat{\tau}^Y \in \mathbb{R}^d$
6: Initialize $\hat{\tau}^X = 0 \in \mathbb{R}^d$
7: **for** $t = 1, \ldots, k$ **do**
8:      Set $\Delta_\infty = \frac{3}{2} + \frac{3}{2} \cdot \mathbb{1}_{t>1}$
9:      Set $s_t = \mathsf{Peel}\left( \hat{\tau}^Y + \frac{\hat{\tau}^X}{t-1}, 1, \Delta_\infty, \frac{\varepsilon}{k} \right)$
10:      Expand $S = S \cup s_t$
11:      Update $\hat{\tau}_{s_t}^Y = -\infty$
12:      **for** $j \notin S$ **do**
13:          Update $\hat{\tau}_j^X = \hat{\tau}_j^X - |\hat{\tau}(X_j, X_{s_t})|$
14: Return $S$

---

Pseudocode for DPKendall appears in Algorithm 1. Its runtime and privacy are easy to verify.

**Theorem 3.4.** DPKendall *runs in time* $O(dkn \log(n))$ *and satisfies* $\varepsilon$-*DP.*

*Proof.* By Lemma 3.3, each computation of Kendall rank correlation takes time $O(n \log(n))$, so Line 2's loop takes time $O(dn \log(n))$, as does each execution of Line 12's loop. Each call to Peel requires $O(d)$ samples of Gumbel noise and thus contributes $O(dk)$ time overall. The loop in Line 7 therefore takes time $O(dkn \log(n))$. The privacy guarantee follows from Lemmas 2.4 and 3.2. $\square$

For comparison, standard OLS on $n$ samples of data with $d$ features requires time $O(d^2 n)$; DPKendall is asymptotically no slower as long as $n \leq O(2^{d/k})$. Since we typically take $k \ll d$, DPKendall is computationally "free" in many realistic data settings.

### 3.3 Utility Guarantee

The proof of DPKendall's utility guarantee combines results about population Kendall rank correlation (Lemma 3.5), empirical Kendall rank correlation concentration (Lemma 3.6), and the accuracy of Peel (Lemma 3.7). The final guarantee (Theorem 3.8) demonstrates that DPKendall selects useful features even in the presence of redundant features.

We start with the population Kendall rank correlation guarantee. Its proof, and all proofs for uncited results in this section, appears in Section 7 in the Appendix.

**Lemma 3.5.** *Suppose that $X_1, \ldots, X_k$ are independent random variables where $X_j \sim N(\mu_j, \sigma_j^2)$. Let $\xi \sim N(0, \sigma_e^2)$ be independent noise. Then if the label is generated by $Y = \sum_{j=1}^k \beta_j X_j + \xi$, for any $j^* \in [k]$,*

$$\tau(X_{j^*}, Y) = \frac{2}{\pi} \cdot \arctan \frac{\beta_{j^*} \sigma_{j^*}}{\sqrt{\sum_{j \neq j^*} \beta_j^2 \sigma_j^2 + \sigma_e^2}}.$$

To interpret this result, recall that $\tau \in [-1, 1]$ and $\arctan$ has domain $\mathbb{R}$, is odd, and has $\lim_{x \to \infty} \arctan x = \pi/2$. Lemma 3.5 thus says that if we fix the other $\sigma_j$ and $\sigma_e$ and take $\sigma_{j^*} \to \infty$, $\tau(X_{j^*}, Y) \to \mathsf{sign}(\beta_{j^*})$ as expected. The next step is to verify that $\hat{\tau}$ concentrates around $\tau$.

**Lemma 3.6** (Lemma 1 [3]). *Given $n$ observations each of random variables $X$ and $Y$, with probability $1 - \eta$,*

$$|\hat{\tau}(X, Y) - \frac{n}{2} \cdot \tau(X, Y)| \leq \sqrt{8n \ln(2/\eta)}.$$

Finally, we state a basic accuracy result for the Gumbel noise employed by Peel.

**Lemma 3.7.** *Given i.i.d. random variables $X_1, \ldots, X_d \sim \mathsf{Gumbel}(b)$, with probability $1 - \eta$,*

$$\max_{j \in [d]} |X_j| \leq b \ln\left(\frac{2d}{\eta}\right).$$

We now have the tools necessary for the final result.

**Theorem 3.8.** *Suppose that $X_1, \ldots, X_k$ are independent random variables where each $X_j \sim N(\mu_j, \sigma_j^2)$. Suppose additionally that of the remaining $d - k$ random variables, for each $j \in [k]$, $n_j$ are copies of $X_j$, where $\sum_{j=1}^k n_j \leq d - k$. For each $j \in [k]$, let $S_j$ denote the set of indices consisting of $j$ and the indices of its copies. Then if the label is generated by $Y = \sum_{j=1}^k \beta_j X_j + \xi$ where $\xi \sim N(0, \sigma_e^2)$ is independent random noise, if*

$$n = \Omega\left(\frac{k \cdot \ln(dk/\eta)}{\varepsilon \cdot \min_{j^* \in [k]}\left\{\left|\arctan \frac{\beta_{j^*} \sigma_{j^*}}{\sqrt{\sum_{j \neq j^*} \beta_j^2 \sigma_j^2 + \sigma_e^2}}\right|\right\}}\right),$$

*then with probability $1 - O(\eta)$, DPKendall correctly selects exactly one index from each of $S_1, \ldots, S_k$.*

*Proof.* The proof reduces to applying the preceding lemmas with appropriate union bounds. Dropping the constant scaling of $\eta$ for neatness, with probability $1 - O(\eta)$:

**1.** Feature-label correlations are large for informative features and small for uninformative features: by Lemma 3.5 and Lemma 3.6, each feature in $j^* \in \cup_{j \in [k]} S_j$ has

$$\hat{\tau}(X_{j^*}, Y) \geq \frac{n}{\pi} \cdot \arctan \frac{\beta_{j^*} \sigma_{j^*}}{\sqrt{\sum_{j \neq j^*} \beta_j^2 \sigma_j^2 + \sigma_e^2}} - \sqrt{8n \ln(d/\eta))}$$

and any $j^* \notin \cup_{j \in [k]} S_j$ has $\hat{\tau}(X_{j^*}, Y) \leq \sqrt{8n \ln(d/\eta)}$.

**2.** Feature-feature correlations are large between copies of a feature and small between independent features: by Lemma 3.5 and Lemma 3.6, for any $j \in [k]$ and $j_1, j_2 \in S_j$,

$$\hat{\tau}(X_{j_1}, X_{j_2}) \geq \frac{n}{2} - \sqrt{8n \ln(d/\eta)}$$

and for any $j_1, j_2$ such that there exists no $S_j$ containing both, $\hat{\tau}(X_{j_1}, X_{j_2}) \leq \sqrt{8n \ln(d/\eta)}$.

**3.** The at most $dk$ draws of Gumbel noise have absolute value bounded by $\frac{k}{\varepsilon} \ln\left(\frac{dk}{\eta}\right)$.

Combining these results, to ensure that DPKendall's $k$ calls to Peel produce exactly one index from each of $S_1, \ldots, S_k$, it suffices to have

$$n \cdot \min_{j^* \in [k]}\left\{\left|\arctan \frac{\beta_{j^*} \sigma_{j^*}}{\sqrt{\sum_{j \neq j^*} \beta_j^2 \sigma_j^2 + \sigma_e^2}}\right|\right\} = \Omega\left([\sqrt{n} + \frac{k}{\varepsilon}] \ln(\frac{dk}{\eta})\right)$$

which rearranges to yield the claim. □

## 4 Experiments

This section collects experimental evaluations of DPKendall and other methods on 25 of the 33 datasets[5]. Descriptions of the relevant algorithms appear in Section 4.1 and Section 4.2. Section 4.3 discusses the results. Experiment code may be found on Github [14].

---

[5]We omit the datasets with OpenML task IDs 361080-361084 as they are restricted versions of other included datasets. We also exclude 361090 and 361097 as non-private OLS obtained $R^2 \ll 0$ on both. Details for the remaining datasets appear in Figure 3 in the Appendix.

## 4.1 Feature Selection Baseline

Our experiments use SubLasso [21] as a baseline "plug-and-play" private feature selection method. At a high level, the algorithm randomly partitions its data into $m$ subsets, computes a non-private Lasso regression model on each, and then privately aggregates these models to select $k$ significant features. The private aggregation process is simple; for each subset's learned model, choose the $k$ features with largest absolute coefficient, then apply private top-$k$ to compute the $k$ features most selected by the $m$ models. Kifer et al. [21] introduced and analyzed SubLasso; we collect its relevant properties in Lemma 4.1. Pseudocode appears in Algorithm 2.

---

**Algorithm 2** SubLasso$(D, k, m, \varepsilon)$

---

1: **Input:** Examples $D = \{(X_i, Y_i)\}_{i=1}^n$, number of selected features $k$, number of models $m$, privacy parameter $\varepsilon$
2: Randomly partition $D$ into $m$ equal-size subsets $S_1, \ldots, S_m$
3: **for** $i = 1, \ldots, m$ **do**
4:     Compute Lasso model $\theta_i$ on $S_i$
5:     Compute set $C_i$ of the $k$ indices of $\theta_i$ with largest absolute value
6:     Compute binary vector $v_i \in \{0,1\}^d$ where $v_{i,j} = \mathbb{1}_{j \in C_i}$
7: Compute $V \in \mathbb{R}^d = \sum_{i=1}^n v_i$
8: Return Peel$(V, k, 1, \varepsilon)$

---

**Lemma 4.1.** SubLasso *is $\varepsilon$-DP and runs in time $O(d^2 n)$.*

*Proof.* The privacy guarantee is immediate from that of Peel (Lemma 2.4). Solving Lasso on $n/m$ data points with $d$-dimensional features takes time $O(\frac{d^2 n}{m})$ [12]. Multiplying through by $m$ produces the final result, since Peel only takes time $O(dk)$. $\square$

Finally, we briefly discuss the role of the intercept feature in SubLasso. As with all algorithms in our experiments, we add an intercept feature (with a constant value of 1) to each vector of features. Each Lasso model is trained on data with this intercept. However, the intercept is removed before the private voting step, $k$ features are chosen from the remaining features, and the intercept is added back afterward. This ensures that privacy is not wasted on the intercept feature, which we always include.

## 4.2 Comparison Algorithms

We evaluate seven algorithms:

1. NonDP is a non-private baseline running generic ordinary least-squares regression.
2. BAS runs Boosted AdaSSP [30] without feature selection. We imitate the parameter settings used by Tang et al. [30] and set feature and gradient clipping norms to 1 and the number of boosting rounds to 100 throughout.
3. Tukey runs the Tukey mechanism [2] without feature selection. We introduce and use a tighter version of the propose-test-release (PTR) check given by Amin et al. [2]. This reduces the number of models needed for PTR to pass. The proof appears in Section 9 in the Appendix and may be of independent interest. To privately choose the number of models $m$ used by the Tukey mechanism, we first privately estimate a $1 - \eta$ probability lower bound on the number of points $n$ using the Laplace CDF, $\tilde{n} = n + \mathsf{Lap}\left(\frac{1}{\varepsilon'}\right) - \frac{\ln(1/2\eta)}{\varepsilon'}$, and then set the number of models to $m = \lfloor \tilde{n}/d \rfloor$. Tukey spends 5% of its $\varepsilon$ privacy budget estimating $m$ and the remainder on the Tukey mechanism.
4. L-BAS runs spends 5% of its $\varepsilon$ privacy budget choosing $m = \lfloor \tilde{n}/k \rfloor$ for SubLasso, 5% of $\varepsilon$ running SubLasso, and then spends the remainder to run BAS on the selected features.
5. L-Tukey spends 5% of its $\varepsilon$ privacy budget choosing $m = \lfloor \tilde{n}/k \rfloor$ for SubLasso, 5% running SubLasso, and the remainder running Tukey on the selected features using the same $m$.
6. K-BAS spends 5% of its $\varepsilon$ privacy budget running DPKendall and then spends the remainder running BAS on the selected features.
7. K-Tukey spends 5% of its $\varepsilon$ privacy budget choosing $m = \lfloor \tilde{n}/k \rfloor$, 5% running DPKendall, and then spends the remainder running the Tukey mechanism on the selected features.

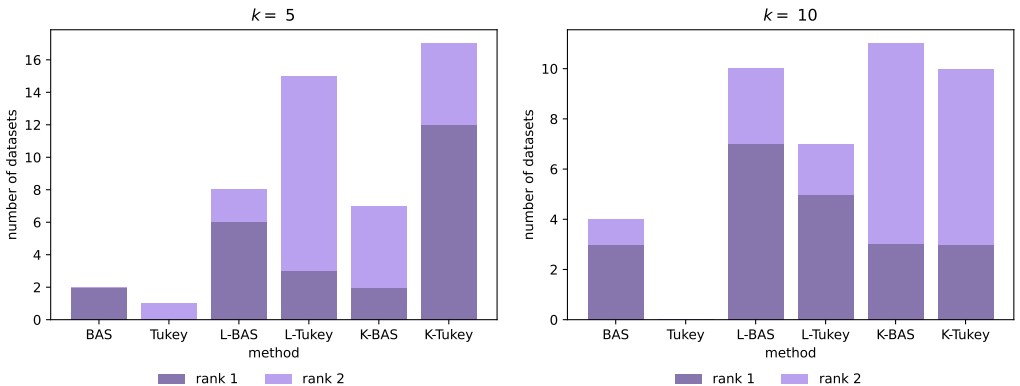

Figure 1: Plots of rank data for each private method.

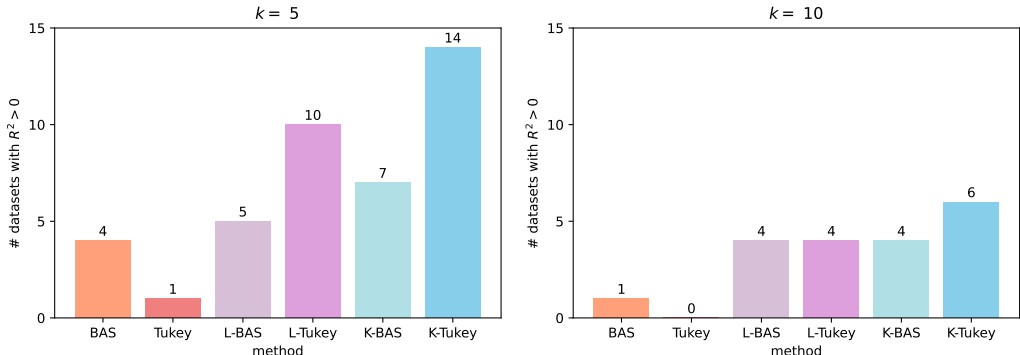

Figure 2: Plots of the number of datasets with positive $R^2$ for each private method.

### 4.3 Results

All experiments use $(\ln(3), 10^{-5})$-DP. Where applicable, 5% of the privacy budget is spent on private feature selection, 5% on choosing the number of models, and the remainder is spent on private regression. Throughout, we use $\eta = 10^{-4}$ as the failure probability for the lower bound used to choose the number of models. For each algorithm and dataset, we run 10 trials using random 90-10 train-test splits and record the resulting test $R^2$ values. Tables of the results at $k = 5$ and $k = 10$ appear in Section 10 in the Appendix. A condensed presentation appears below. At a high level, we summarize the results in terms of *relative* and *absolute* performance.

#### 4.3.1 Relative Performance

First, for each dataset and method, we compute the median $R^2$ of the final model across the 10 trials and then rank the methods, with the best method receiving a rank of 1. Figure 1 plots the number of times each method is ranked first or second.

At $k = 5$ (left), K-Tukey performs best by a significant margin: it ranks first on 48% of datasets, twice the fraction of any other method. It also ranks first or second on the largest fraction of datasets (68%). In some contrast, L-BAS obtains the top ranking on 24% of datasets, whereas K-BAS only does so on 8%; nonetheless, the two have nearly the same number of total first or second rankings. At $k = 10$ (right), no clear winner emerges among the feature selecting methods, as L-BAS, K-BAS, and K-Tukey are all first or second on around half the datasets[6], though the methods using SubLasso have a higher share of datasets ranked first.

---

[6]Note that $k = 10$ only uses 21 datasets. This is because 4 of the 25 datasets used for $k = 5$ have $d < 10$.

### 4.3.2 Absolute Performance

Second, we count the number of datasets on which the method achieves a positive median $R^2$ (Figure 2), recalling that $R^2 = 0$ is achieved by the trivial model that always predicts the mean label. The $k = 5$ (left) setting again demonstrates clear trends: K-Tukey attains $R^2 > 0$ on 56% of datasets, L-Tukey does so on 40%, K-BAS does so on 28%, and L-BAS on 20%. DPKendall therefore consistently demonstrates stronger performance than SubLasso. As in the rank data, at $k = 10$ the picture is less clear. However, K-Tukey is still best by some margin, with the remaining feature selecting methods all performing roughly equally well.

### 4.4 Discussion

A few trends are apparent from Section 4.3. First, DPKendall generally achieves stronger final utility than SubLasso, particularly for the Tukey mechanism; the effect is similar but smaller for $k = 10$; and feature selection generally improves the performance of private linear regression.

**Comparing Feature Selection Algorithms**. A possible reason for DPKendall's improvement over SubLasso is that, while SubLasso takes advantage of the stability properties that Lasso exhibits in certain data regimes [21], this stability does not always hold in practice. Another possible explanation is that the feature coefficients passed to Peel scale with $O(n)$ for DPKendall and $m = O(\frac{n}{d})$ or $m = O(\frac{n}{k})$ for SubLasso. Both algorithm's invocations of Peel add noise scaling with $O(\frac{k}{\varepsilon})$, so DPKendall's larger scale makes it more robust to privacy-preserving noise. Finally, we emphasize that DPKendall achieves this even though its $O(dkn \log(n))$ runtime is asymptotically smaller than the $O(d^2 n)$ runtime of SubLasso in most settings.

**Choosing k**. Next, we examine the decrease in performance from $k = 5$ to $k = 10$. Conceptually, past a certain point adding marginally less informative features to a private model may worsen utility due to the privacy cost of considering these features. Moving from $k = 5$ to $k = 10$ may cross this threshold for many of our datasets; note from Figure 4 and Figure 5 in the Appendix that, of the 21 datasets used for $k = 10$, 86% witness their highest private $R^2$ in the $k = 5$ setting[7]. Moreover, from $k = 5$ to $k = 10$ the total number of positive $R^2$ datasets across methods declines by more than 50%, from 41 to 19, with all methods achieving positive $R^2$ less frequently at $k = 10$ than $k = 5$. We therefore suggest $k = 5$ as the more relevant setting, and a good choice in practice.

**The Effect of Private Feature Selection**. Much work aims to circumvent generic lower bounds for privately answering queries by taking advantage of instance-specific structure [6, 16, 33, 5]. Similar works exist for private optimization, either by explicitly incorporating problem information [35, 18] or showing that problem-agnostic algorithms can, under certain conditions, take advantage of problem structure organically [24]. We suggest that this paper makes a similar contribution: feature selection reduces the need for algorithms like Boosted AdaSSP and the Tukey mechanism to "waste" privacy on computations over irrelevant features. This enables them to apply less obscuring noise to the signal contained in the selected features. The result is the significant increase in utility shown here.

## 5 Conclusion

We briefly discuss DPKendall's limitations. First, it requires an end user to choose the number of features $k$ to select; we suggest $k = 5$ as a reasonable first cut. Second, DPKendall's use of Kendall rank correlation may struggle when ties are intrinsic to the data's structure, e.g., when the data is categorical, as a monotonic relationship between feature and label becomes less applicable. Finally, Kendall rank correlation may fail to distinguish between linear and nonlinear monotonic feature-label relationships, even though the former is more likely to be useful for linear regression. Unfortunately, it is not obvious how to incorporate relationships more sophisticated than simple monotonicity without sacrificing rank correlation's low sensitivity. Answering these questions may be an interesting avenue for future work.

Nonetheless, the results of this paper demonstrate that DPKendall expands the applicability of plug-and-play private linear regression algorithms while providing more utility in less time than the current state of the art. We therefore suggest that DPKendall presents a step forward for practical private linear regression.

---

[7]The exceptions are datasets 361075, 361091, and 361103.

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

# 6 Alternative Notions of Correlation

## 6.1 Pearson

The Pearson correlation between some feature $X$ and the label $Y$ is defined by

$$r(X, Y) := \frac{\mathsf{Cov}\,(X, Y)}{\sqrt{\mathsf{Var}\,(X)\,\mathsf{Var}\,(Y)}}. \tag{2}$$

Evaluated on a sample of $n$ points, this becomes

$$r(X, Y) := \frac{\sum_{i=1}^{n}(X_i - \bar{X})(Y_i - \bar{Y})}{\sqrt{\sum_{i=1}^{n}(X_i - \bar{X})^2}\sqrt{\sum_{i=1}^{n}(Y_i - \bar{Y})^2}}.$$

where $\bar{X}$ and $\bar{Y}$ are sample means.

**Lemma 6.1.** $r \in [-1, 1]$.

*Proof.* This is immediate from Cauchy-Schwarz. $\qquad\square$

Note that a value of $-1$ is perfect anticorrelation and a value of $1$ is perfect correlation. A downside of Pearson correlation is that it is not robust. In particular, its sensitivity is the same as its range.

**Lemma 6.2.** *Pearson correlation has $\ell_1$-sensitivity $\Delta(r) = 2$.*

*Proof.* Consider neighboring databases $D = \{(-1, -1), (1, 1)\}$ and $D' = \{(-1, -1), (1, 1), (-c, c)\}$. $r(D) = 1$, but

$$r(D') = \frac{(-1 + \frac{c}{3})(-1 - \frac{c}{3}) + (1 + \frac{c}{3})(1 - \frac{c}{3}) + (-\frac{2c}{3} \cdot \frac{2c}{3})}{\sqrt{(-1 + \frac{c}{3})^2 + (1 + \frac{c}{3})^2 + (-\frac{2c}{3})^2}\sqrt{(-1 - \frac{c}{3})^2 + (1 - \frac{c}{3})^2 + (\frac{2c}{3})^2}}$$

$$\approx \frac{-(\frac{2c}{3})^2}{(\frac{2c}{3})^2} = -1$$

where the approximation is increasingly accurate as $c$ grows. $\qquad\square$

## 6.2 Spearman

Spearman rank correlation is Pearson correlation applied to rank variables.

**Definition 6.3.** *Given data points $X_1, \ldots, X_n \in \mathbb{R}$, the corresponding* rank variables *$R(X_1), \ldots, R(X_n)$ are defined by setting $R(X_i)$ to the position of $X_i$ when $X_1, \ldots, X_n$ are sorted in descending order. Given data $(X, Y)$, the* Spearman rank correlation *is $\rho(X, Y) := r(R(X), R(Y))$.*

For example, given database $D = \{(0, 1), (1, 0), (2, 3)\}$, its Spearman rank correlation is

$$\rho(D) := r(\{(3, 2), (2, 3), (1, 1)\}).$$

A useful privacy property of rank is that it does not depend on data scale. Moreover, if there are no ties then Spearman rank correlation admits a simple closed form.

**Lemma 6.4.** $\rho(X, Y) = 1 - \frac{6\sum_{i=1}^{n}(R(X_i) - R(Y_i))^2}{n(n^2 - 1)}$.

If we consider adding a "perfectly unsorted" data point with rank variables $(1, n + 1)$ to a perfectly sorted database with rank variables $\{(1, 1), (2, 2), \ldots, (n, n)\}$, $\rho$ changes from 1 to $1 - \frac{6(n + n^2)}{n(n^2 - 1)}$. The sensitivity's dependence on $n$ complicates its usage with add-remove privacy. Nonetheless, both Spearman and Kendall correlation's use of rank makes them relatively easy to compute privately, and as the two methods are often used interchangeably in practice, we opt for Kendall rank correlation for simplicity.

# 7 Deferred Proofs From Section 3.3

We first restate and prove Lemma 3.5.

**Lemma 3.5.** *Suppose that $X_1, \ldots, X_k$ are independent random variables where $X_j \sim N(\mu_j, \sigma_j^2)$. Let $\xi \sim N(0, \sigma_e^2)$ be independent noise. Then if the label is generated by $Y = \sum_{j=1}^k \beta_j X_j + \xi$, for any $j^* \in [k]$,*

$$\tau(X_{j^*}, Y) = \frac{2}{\pi} \cdot \arctan \frac{\beta_{j^*} \sigma_{j^*}}{\sqrt{\sum_{j \neq j^*} \beta_j^2 \sigma_j^2 + \sigma_e^2}}.$$

*Proof.* Recall from Definition 3.1 the population formulation of $\tau$,

$$\tau(X, Y) = \mathbb{P}\left[(X - X')(Y - Y') > 0\right] - \mathbb{P}\left[(X - X')(Y - Y') < 0\right] \tag{3}$$

where $X$ and $X'$ are i.i.d., as are $Y$ and $Y'$. In our case, we can define $Z = \sum_{j \neq j^*} \beta_j X_j + \xi$ and rewrite the first term of Equation 3 for our setting as

$$\int_0^\infty f_{X_{j^*} - X'_{j^*}}(t) \cdot [1 - F_{Z - Z'}(-\beta_{j^*}t)]dt + \int_{-\infty}^0 f_{X_{j^*} - X'_{j^*}}(t) \cdot F_{Z - Z'}(-\beta_{j^*}t)dt. \tag{4}$$

where $f_A$ and $F_A$ denote densities and cumulative distribution functions of random variable $A$, respectively. Since the relevant distributions are all Gaussian, $X_{j^*} - X'_{j^*} \sim N(0, 2\sigma_{j^*}^2)$ and $Z - Z' \sim N(0, 2[\sum_{j \neq j^*} \beta_j^2 \sigma_j^2 + \sigma_e^2])$. For neatness, shorthand $\sigma^2 = 2\sigma_{j^*}^2$ and $\sigma_{-1}^2 = 2[\sum_{j \neq j^*} \beta_j^2 \sigma_j^2 + \sigma_e^2]$. Then if we let $\phi$ denote the PDF of a standard Gaussian and $\Phi$ the CDF of a standard Gaussian, Equation 4 becomes

$$\int_0^\infty \frac{1}{\sigma} \phi(t/\sigma) \cdot [1 - \Phi(-\beta_{j^*}t/\sigma_{-1})]dt + \int_{-\infty}^0 \frac{1}{\sigma}\phi(t/\sigma)\Phi(-\beta_{j^*}t/\sigma_{-1})dt$$

$$= \frac{1}{\sigma}\left[\int_0^\infty \phi(t/\sigma)\Phi(\beta_{j^*}t/\sigma_{-1})dt + \int_{-\infty}^0 \phi(-t/\sigma)\Phi(-\beta_{j^*}t/\sigma_{-1})dt\right]$$

$$= \frac{2}{\sigma}\int_0^\infty \phi(t/\sigma)\Phi(\beta_{j^*}t/\sigma_{-1})dt.$$

We can similarly analyze the second term of Equation 3 to get

$$\int_0^\infty f_{X_{j^*} - X'_{j^*}}(t) \cdot F_{Z - Z'}(-\beta_{j^*}t)dt + \int_{-\infty}^0 f_{X_{j^*} - X'_{j^*}}(t) \cdot [1 - F_{Z - Z'}(-\beta_{j^*}t)]dt$$

$$= \int_0^\infty \frac{1}{\sigma}\phi(t/\sigma)\Phi(-\beta_{j^*}t/\sigma_{-1})dt + \int_{-\infty}^0 \frac{1}{\sigma}\phi(t/\sigma)\Phi(\beta_{j^*}t/\sigma_{-1})dt$$

$$= \frac{2}{\sigma}\int_{-\infty}^0 \phi(t/\sigma)\Phi(\beta_{j^*}t/\sigma_{-1})dt.$$

Using both results, we get

$$\tau(X_1, Y) = \frac{2}{\sigma}\left[\int_0^\infty \phi(t/\sigma)\Phi(\beta_{j^*}t/\sigma_{-1})dt - \int_{-\infty}^0 \phi(t/\sigma)\Phi(\beta_{j^*}t/\sigma_{-1})dt\right]$$

$$= \frac{2}{\sigma}\left[\int_0^\infty \phi(t/\sigma)\Phi(\beta_{j^*}t/\sigma_{-1})dt - \int_0^\infty \phi(t/\sigma)[1 - \Phi(\beta_{j^*}t/\sigma_{-1})]dt\right]$$

$$= \frac{4}{\sigma}\int_0^\infty \phi(t/\sigma)\Phi(\beta_{j^*}t/\sigma_{-1})dt - 1$$

$$= \frac{4}{\sigma} \cdot \frac{\sigma}{2\pi}\left(\frac{\pi}{2} + \arctan\frac{\beta_{j^*}\sigma}{\sigma_{-1}}\right) - 1$$

$$= \frac{2}{\pi} \cdot \arctan\frac{\beta_{j^*}\sigma}{\sigma_{-1}}.$$

where the third equality uses $\int_0^\infty \phi(t/\sigma)dt = \frac{\sigma}{2}$ and the fourth equality comes from Equation 1,010.4 of Owen [26]. Substituting in the values of $\sigma$ and $\sigma_{-1}$ yields the claim. $\square$

Next is Lemma 3.7.

**Lemma 3.7.** *Given i.i.d. random variables $X_1, \ldots, X_d \sim$ Gumbel $(b)$, with probability $1 - \eta$,*

$$\max_{j \in [d]} |X_j| \leq b \ln \left( \frac{2d}{\eta} \right).$$

*Proof.* Recall from Definition 2.3 that Gumbel $(b)$ has density $f(x) = \frac{1}{b} \cdot \exp\left( -\frac{x}{b} - e^{-x/b} \right)$. Then $\frac{f(x)}{f(-x)} = \exp\left( -\frac{x}{b} - e^{-x/b} - \frac{x}{b} + e^{x/b} \right)$. For $z \geq 0$, by $e^z = \sum_{n=0}^{\infty} \frac{z^n}{n!}$,

$$2z + e^{-z} \leq 2z + 1 - z + \frac{z^2}{2} = 1 + z + \frac{z^2}{2} \leq e^z$$

so since $b \geq 0$, $f(x) \geq f(-x)$ for $x \geq 0$. Letting $F(z) = \exp(-\exp(-z/b))$ denote the CDF for Gumbel $(b)$, it follows that for $z \geq 0$, $1 - F(z) \geq F(-z)$. Thus the probability that $\max_{j \in [d]} |X_j|$ exceeds $t$ is upper bounded by $2d(1 - F(t))$. The claim then follows from rearranging the inequality

$$2d(1 - F(t)) \leq \eta$$
$$1 - \frac{\eta}{2d} \leq F(t)$$
$$1 - \frac{\eta}{2d} \leq \exp(-\exp(-t/b))$$
$$\ln\left( 1 - \frac{\eta}{2d} \right) \leq -\exp(-t/b)$$
$$-b \ln\left( -\ln\left( 1 - \frac{\eta}{2d} \right) \right) \leq t$$
$$b \ln\left( \frac{1}{-\ln(1 - \eta/[2d])} \right) \leq t$$

and using $-\ln(1 - x) = \sum_{i=1}^{\infty} x^i / i \geq x$. $\qquad \square$

## 8 Datasets

A summary of the datasets used in our experiments appears in Figure 3.

We briefly discuss the role of the intercept in these datasets. Throughout, we explicitly add an intercept feature (constant 1) to each vector. Where feature selection is applied, we explicitly remove the intercept feature during feature selection and then add it back to the $k$ selected features afterward. The resulting regression problem therefore has dimension $k + 1$. We do this to avoid spending privacy budget selecting the intercept feature.

## 9 Modified PTR Lemma

This section describes a simple tightening of Lemma 3.6 from Amin et al. [2] (which is itself a small modification of Lemma 3.8 from Brown et al. [7]). Tukey uses the result as its propose-test-release (PTR) check, so tightening it makes the check easier to pass. Proving the result will require introducing details of the Tukey algorithm. The following exposition aims to keep this document both self-contained and brief; the interested reader should consult the expanded treatment given by Amin et al. [2] for further details.

Tukey depth was introduced by Tukey [32]. Amin et al. [2] used an approximation for efficiency. Roughly, Tukey depth is a notion of depth for a collection of points in space. (Exact) Tukey depth is evaluated over all possible directions in $\mathbb{R}^d$, while approximate Tukey depth is evaluated only over axis-aligned directions.

**Definition 9.1** ([32, 2]). *A halfspace $h_v$ is defined by a vector $v \in \mathbb{R}^d$, $h_v = \{ y \in \mathbb{R}^d \mid \langle v, y \rangle \geq 0 \}$. Let $E = \{ e_1, ..., e_d \}$ be the canonical basis for $\mathbb{R}^d$ and let $D \subset \mathbb{R}^d$. The* approximate Tukey depth *of*

| OpenML Task ID | n | d | $\lfloor n/d \rfloor$ | $\lfloor n/5 \rfloor$ | $\lfloor n/10 \rfloor$ |
|---|---|---|---|---|---|
| 361072 | 8192 | 22 | 372 | 1638 | 819 |
| 361073 | 15000 | 27 | 555 | 3000 | 1500 |
| 361074 | 16599 | 17 | 976 | 3319 | 1659 |
| 361075 | 7797 | 614 | 12 | 1559 | 779 |
| 361076 | 6497 | 12 | 541 | 1299 | 649 |
| 361077 | 13750 | 34 | 404 | 2750 | 1375 |
| 361078 | 20640 | 9 | 2293 | 4128 | 2064 |
| 361079 | 22784 | 17 | 1340 | 4556 | 2278 |
| 361085 | 10081 | 7 | 1440 | 2016 | 1008 |
| 361087 | 13932 | 14 | 995 | 2786 | 1393 |
| 361088 | 21263 | 80 | 265 | 4252 | 2126 |
| 361089 | 20640 | 9 | 2293 | 4128 | 2064 |
| 361091 | 515345 | 91 | 5663 | 103069 | 51534 |
| 361092 | 8885 | 83 | 107 | 1777 | 888 |
| 361093 | 4052 | 13 | 311 | 810 | 405 |
| 361094 | 8641 | 6 | 1440 | 1728 | 864 |
| 361095 | 166821 | 24 | 6950 | 33364 | 16682 |
| 361096 | 53940 | 27 | 1997 | 10788 | 5394 |
| 361098 | 10692 | 18 | 594 | 2138 | 1069 |
| 361099 | 17379 | 21 | 827 | 3475 | 1737 |
| 361100 | 39644 | 74 | 535 | 7928 | 3964 |
| 361101 | 581835 | 32 | 18182 | 116367 | 58183 |
| 361102 | 21613 | 20 | 1080 | 4322 | 2161 |
| 361103 | 394299 | 27 | 14603 | 78859 | 39429 |
| 361104 | 241600 | 16 | 15100 | 48320 | 24160 |

Figure 3: Parameters of the 25 datasets used in our experiments.

a point $y \in \mathbb{R}^d$ with respect to D, denoted $\tilde{T}_D(y)$, is the minimum number of points in D in any of the $2d$ halfspaces determined by E containing y,

$$\tilde{T}_D(y) = \min_{v \in \pm E \ s.t. \ y \in h_v} \sum_{x \in D} \mathbb{1}_{x \in h_v}.$$

At a high level, Lemma 3.6 from Amin et al. [2] is a statement about the volumes of regions of different depths. The next step is to formally define these volumes.

**Definition 9.2** ([2]). *Given database D, define $S_{i,D} = \{y \in \mathbb{R}^d \mid \tilde{T}_D(y) \geq i\}$ to be the set of points with approximate Tukey depth at least i in D and $V_{i,D} = \mathsf{vol}(S_{i,D})$ to be the volume of that set. When D is clear from context, we write $S_i$ and $V_i$ for brevity. We also use $w_D(V_{d,D}) :=  \int_{S_{d,D}} \exp(\varepsilon \cdot \tilde{T}_D(y)) dy$ to denote the weight assigned to $V_{d,D}$ by an exponential mechanism whose score function is $\tilde{T}_D$.*

Amin et al. [2] define a family of mechanisms $A_1, A_2, \ldots$, where $A_t$ runs the exponential mechanism to choose a point of approximately maximal Tukey depth, but restricted to the domain of points with Tukey depth at least $t$. Since this domain is a data-dependent quantity, they use the PTR framework to select a safe depth $t$. We briefly recall the definitions of "safe" and "unsafe" databases given by Brown et al. [7], together with the key PTR result from Amin et al. [2].

**Definition 9.3** (Definitions 2.1 and 3.1 [7]). *Two distributions $\mathcal{P}, \mathcal{Q}$ over domain $\mathcal{W}$ are $(\varepsilon, \delta)$-indistinguishable, denoted $\mathcal{P} \approx_{\varepsilon,\delta} \mathcal{Q}$, if for any measurable subset $W \subset \mathcal{W}$,*

$$\mathbb{P}_{w \sim \mathcal{P}}[w \in W] \leq e^\varepsilon \mathbb{P}_{w \sim \mathcal{Q}}[w \in W] + \delta \text{ and } \mathbb{P}_{w \sim \mathcal{Q}}[w \in W] \leq e^\varepsilon \mathbb{P}_{w \sim \mathcal{P}}[w \in W] + \delta.$$

*Database D is $(\varepsilon, \delta, t)$-safe if for all neighboring $D' \sim D$, we have $A_t(D) \approx_{\varepsilon,\delta} A_t(D')$. Let $\mathsf{Safe}_{(\varepsilon,\delta,t)}$ be the set of safe databases, and let $\mathsf{Unsafe}_{(\varepsilon,\delta,t)}$ be its complement.*

We can now restate Lemma 3.6 from Amin et al. [2]. Informally, it states that if the volume of an "outer" region of Tukey depth is not much larger than the volume of an "inner" region, the difference

in depth between the two is a lower bound on the distance to an unsafe database. For the purpose of this paper, Tukey applies this result by finding such a $k$, adding noise for privacy, and checking that the resulting distance to unsafety is large enough that subsequent steps will be privacy-safe.

**Lemma 9.4.** *Define $M(D)$ to be a mechanism that receives as input database $D$ and computes the largest $k \in \{0, \ldots, t-1\}$ such that there exists $g > 0$ where*

$$\frac{V_{t-k-1,D}}{V_{t+k+g+1,D}} \cdot e^{-\varepsilon g/2} \leq \delta$$

*or outputs $-1$ if the inequality does not hold for any such $k$. Then for arbitrary $D$*

1. *$M$ is 1-sensitive, and*

2. *for all $z \in \mathtt{Unsafe}_{(\varepsilon, 4e^\varepsilon \delta, t)}$, $d_H(D, z) > M(D)$.*

We provide a drop-in replacement for Lemma 9.4 that slightly weakens the requirement placed on $k$.

**Lemma 9.5.** *Define $M(D)$ to be a mechanism that receives as input database $D$ and computes the largest $k \in \{0, \ldots, t-1\}$ such that*

$$\frac{V_{t-k-1,D}}{w_D(V_{t+k-1,D})} \cdot e^{\varepsilon(t+k+1)} \leq \delta$$

*or outputs $-1$ if the inequality does not hold for any such $k$. Then for arbitrary $D$*

1. *$M$ is 1-sensitive, and*

2. *for all $z \in \mathtt{Unsafe}_{(\varepsilon, 4e^\varepsilon \delta, t)}$, $d_H(D, z) > M(D)$.*

The new result therefore replaces the denominator $V_{t+k+g+1,D} \cdot e^{\varepsilon g/2}$ with denominator $\frac{w_D(V_{t+k-1,D})}{e^{\varepsilon(t+k+1)}}$. Every point in $V_{t+k-1,D}$ of depth at least $t + k + g + 1$ has score at least $t + k + g + 1$, so $\frac{w_D(V_{t+k-1,D})}{e^{\varepsilon(t+k+1)}} \geq V_{t+k+g+1,D} \cdot e^{\varepsilon g}$, so $V_{t+k+g+1,D} \cdot e^{\varepsilon g/2} \leq \frac{w_D(V_{t+k+g+1,D})}{e^{\varepsilon(t+k+1)}}$ and the check for the new result is no harder to pass. To see that it may be easier, note that only the new result takes advantage of the higher scores of deeper points in $V_{t+k-1,D}$.

*Proof of Lemma 9.5.* First we prove item 1. Let $D$ and $D'$ be any neighboring databases and suppose WLOG that $D' = D \cup \{x\}$. We want to show that $|M(D) - M(D')| \leq 1$.

First we prove relationships between the points with approximate Tukey depth at least $p$ and $p - 1$ in datasets $D$ and $D'$. From the definition of approximate Tukey depth, together with the fact that $D'$ contains one additional point, for any point $y$, we are guaranteed that $\tilde{T}_D(y) \leq \tilde{T}_{D'}(y) \leq \tilde{T}_D(y) + 1$. Recall that $S_{p,D}$ is the set of points with approximate Tukey depth at least $p$ in $D$. This implies that $S_{p+1,D} \subset S_{p,D}$. Next, since for every point $y$ we have $\tilde{T}_{D'}(y) \geq \tilde{T}_D(y)$, we have that $S_{p,D} \subset S_{p,D'}$. Finally, since $\tilde{T}_D(y) \geq \tilde{T}_{D'}(y) - 1$, we have that $S_{p,D'} \subset S_{p-1,D}$. Taken together, we have

$$S_{p+1,D} \subset S_{p,D} \subset S_{p,D'} \subset S_{p-1,D}.$$

It follows that

$$V_{p+1,D} \leq V_{p,D} \leq V_{p,D'} \leq V_{p-1,D}. \tag{5}$$

Next, since the unnormalized exponential mechanism density $y \mapsto \exp(y\hat{T}_D(y))$ is non-negative, we have that $w_D(V_{p+1,D}) = \int_{S_{p+1,D}} \exp(\epsilon \tilde{T}_D(y)) \, dy \leq \int_{S_{p,D}} \exp(\epsilon \tilde{T}_D(y)) \, dy = w_D(V_{p,D})$. Using the fact that $\tilde{T}_D(y) \leq \tilde{T}_{D'}(y)$, we have that $w_D(V_{p,D}) = \int_{S_{p,D}} \exp(\epsilon \tilde{T}_D(y)) \, dy \leq \int_{S_{p,D'}} \exp(\epsilon \tilde{T}_{D'}(y)) \, dy = w_{D'}(V_{p,D'})$. Finally, using the fact that $\tilde{T}_{D'}(y) \leq \tilde{T}_D(y) + 1$, we have $w_{D'}(V_{p,D'}) = \int_{S_{p,D'}} \exp(\varepsilon \tilde{T}_{D'}(y)) \, dy \leq \int_{S_{p-1,D}} \exp(\epsilon \tilde{T}_D(y) + \epsilon) \, dy = e^\epsilon w_D(V_{p-1,D})$. Together, this gives

$$w_D(V_{p+1,D}) \leq w_D(V_{p,D}) \leq w_{D'}(V_{p,D'}) \leq w_D(V_{p-1,D}) \cdot e^\varepsilon. \tag{6}$$

Now suppose there exists $k^*_{D'} \geq 0$ such that $\frac{V_{t-k^*_{D'}-1,D'}}{w_{D'}(V_{t+k^*_{D'}+2,D'})} \cdot e^{\varepsilon(t+k^*_{D'}+1)} \leq \delta$. Then by Equation 5, $V_{t-k^*_{D'}-1,D'} \geq V_{t-k^*_{D'},D}$, and by Equation 6 $w_{D'}(V_{t+k^*_{D'}+2,D'}) \leq w_D(V_{t+k^*_{D'}+1,D}) \cdot e^\varepsilon$, so

$\frac{V_{t-k^*_{D'},D}}{w_D(V_{t+k^*_{D'}+1,D})} \cdot e^{\varepsilon(t+k^*_{D'})} \leq \delta$ and then $k^*_D \geq k^*_{D'} - 1$. Similarly, if there exists $k^*_D \geq 0$ such that

$\frac{V_{t-k^*_D-1,D}}{w_D(V_{t+k^*_D+2,D})} \cdot e^{\varepsilon(t+k^*_D+1)} \leq \delta$, then by Equation 5 $V_{t-k^*_D-1,D} \geq V_{t-k^*_D,D'}$, and by Equation 6

$w_D(V_{t+k^*_D+2,D}) \leq w_{D'}(V_{t+k^*_D+2,D'})$, so $\frac{V_{t-k^*_D,D'}}{w_{D'}(V_{t+k^*_x+2,D'})} \cdot e^{\varepsilon(t+k^*_{D'})} \leq \delta$, and $k^*_{D'} \geq k^*_D - 1$. Thus if $k^*_D \geq 0$ or $k^*_{D'} \geq 0$, $|k^*_D - k^*_{D'}| \leq 1$. The result then follows since $k^* \geq -1$.

As in Lemma 9.4, item 2 is a consequence of Lemma 3.8 from Brown et al. [7] and the fact that $k^* = -1$ is a trivial lower bound on distance. The only change made to the proof of Lemma 3.8 of Brown et al. [7] is, in its notation, to replace its denominator lower bound with

$$w_z(V_{t-1,z}) \geq w_x(V_{t+k-1,x}) \cdot e^{-k\varepsilon}$$

This uses the fact that $x$ and $z$ differ in at most the addition or removal of $k$ data points. Thus $V_{t+k-1,x} \leq V_{t-1,z}$, and no point's score increases by more than $k$ from $V_{t-1,z}$ to $V_{t+k-1,x}$. Since their numerator upper bound is $V_{t-k-1,x} \cdot e^{\varepsilon(t+1)}$ (note that the 2 is dropped here because approximate Tukey depth is monotonic; see Section 7.3 of [2] for details), the result follows. $\square$

## 10 Extended Experiment Results

| Task ID | NonDP | BAS | Tukey | L-BAS | L-Tukey | K-BAS | K-Tukey |
|---------|-------|-----|-------|-------|---------|-------|---------|
| 361072 | 7.3e-01 | **7.8e-01** | $-\infty$ | -2.7e-02 | 1.0e-01 | **2.1e-01** | 1.0e-01 |
| 361073 | 4.6e-01 | -3.4e+00 | $-\infty$ | -4.6e-01 | **-2.3e-02** | -4.8e-01 | **-4.3e-01** |
| 361074 | 8.0e-01 | -4.5e+04 | -6.6e+02 | -8.2e+02 | **-1.7e+02** | -5.6e+05 | **3.1e-01** |
| 361075 | 6.2e-01 | -3.6e+02 | $-\infty$ | 3.2e-03 | **2.2e-02** | 5.2e-03 | **7.1e-02** |
| 361076 | 2.8e-01 | -3.4e+00 | $-\infty$ | -5.6e-01 | **-2.5e-01** | -4.3e+00 | **8.5e-02** |
| 361077 | 8.2e-01 | -2.1e+08 | $-\infty$ | -1.0e+07 | **3.7e-01** | -1.9e+08 | **6.6e-01** |
| 361078 | 6.5e-01 | -1.1e+03 | -6.4e+00 | **-7.4e-01** | -1.2e+00 | -6.1e+02 | **-9.5e-01** |
| 361079 | 2.6e-01 | -5.4e+00 | -1.0e+01 | -1.3e+01 | **-2.e+00** | -9.4e+00 | **-6.8e-01** |
| 361085 | 3.3e-01 | -2.e+01 | **3.3e-01** | -1.7e+01 | 2.8e-01 | -1.3e+01 | **3.7e-01** |
| 361087 | 7.2e-01 | -3.1e+03 | -8.8e+04 | -4.7e+00 | -1.9e+02 | **-2.8e+00** | **6.6e-01** |
| 361088 | 7.3e-01 | 5.4e-02 | $-\infty$ | 1.4e-01 | **3.e-01** | 2.1e-01 | **3.6e-01** |
| 361089 | 6.2e-01 | -3.e+00 | -1.5e+01 | **-1.3e+00** | -5.9e+00 | -5.9e+04 | **-2.6e-01** |
| 361091 | 2.4e-01 | -3.0e+04 | -1.9e+00 | -3.0e+04 | **3.8e-02** | -3.1e+04 | **4.9e-02** |
| 361092 | 2.0e-02 | -4.7e+05 | $-\infty$ | **-4.1e+04** | -1.4e+08 | **-1.3e+05** | -9.3e+08 |
| 361093 | 4.3e-01 | -5.6e-02 | $-\infty$ | **-4.6e-02** | $-\infty$ | **-4.9e-02** | $-\infty$ |
| 361094 | 8.3e-01 | **4.2e-01** | -3.8e+05 | **-1.0e+00** | -4.2e+05 | -2.4e+00 | -3.0e+05 |
| 361095 | 2.2e-01 | -1.0e+00 | -1.2e+09 | -8.0e-01 | **2.e-01** | 1.5e-01 | **2.1e-01** |
| 361096 | 9.7e-01 | -9.1e+02 | -1.e+08 | -4.9e+00 | **9.1e-01** | 7.1e-01 | **8.8e-01** |
| 361098 | 8.6e-01 | -2.9e+01 | $-\infty$ | -3.9e+00 | **-3.1e+00** | **-2.2e-01** | -5.6e+00 |
| 361099 | 4.0e-01 | -4.4e-01 | -4.8e+10 | -4.8e-01 | **-9.8e-03** | -4.2e-01 | **3.2e-01** |
| 361100 | 1.2e-01 | -6.3e+01 | $-\infty$ | -1.1e+00 | **-2.8e-02** | -4.6e-01 | **-1.7e-01** |
| 361101 | 3.3e-01 | -9.3e+01 | -5.9e+08 | **3.6e-01** | 2.e-01 | **3.0e-01** | 2.1e-01 |
| 361102 | 7.6e-01 | -4.1e+02 | -1.4e+15 | **6.8e-02** | **-2.6e-01** | -4.0e+00 | -5.3e+00 |
| 361103 | 5.3e-01 | 4.3e-01 | -6.3e+07 | **5.e-01** | **4.9e-01** | 3.3e-01 | 4.8e-01 |
| 361104 | 6.8e-01 | -3.7e+03 | -6.4e+07 | -1.5e+02 | -2.6e+05 | **-8.3e-01** | **-2.9e+01** |

Figure 4: $R^2$ values for $k = 5$. Each entry is the median test value from 10 trials. The top two private values for each dataset are bolded.

| Task ID | NonDP | BAS | Tukey | L-BAS | L-Tukey | K-BAS | K-Tukey |
|---------|-------|-----|-------|-------|---------|-------|---------|
| 361072 | 7.4e-01 | **5.3e-01** | $-\infty$ | 3.5e-01 | 1.8e-01 | **7.3e-01** | 8.2e-02 |
| 361073 | 4.6e-01 | -8.2e-01 | $-\infty$ | **-4.4e-01** | -1.7e+00 | **-4.8e-01** | -1.6e+01 |
| 361074 | 8.1e-01 | -2.3e+06 | -1.3e+03 | -5.6e+06 | **-3.4e+02** | -7.2e+04 | **-1.4e+02** |
| 361075 | 6.2e-01 | -1.7e+02 | $-\infty$ | **7.5e-02** | -9.7e-01 | **3.1e-02** | -2.8e+00 |
| 361076 | 2.8e-01 | -2.4e+02 | -8.2e+04 | **-2.7e+01** | -3.5e+03 | **-2.4e+01** | -1.4e+03 |
| 361077 | 7.9e-01 | -1.1e+08 | $-\infty$ | -1.4e+07 | **-3.1e+01** | -2.1e+08 | **-1.7e+02** |
| 361079 | 2.4e-01 | -1.8e+00 | -3.7e+01 | -1.2e+01 | **-8.1e-01** | -4.1e+00 | **-1.5e+00** |
| 361087 | 7.1e-01 | -3.7e+01 | -1.5e+04 | **-7.2e+00** | -4.8e+03 | -8.5e+01 | **-2.e+01** |
| 361088 | 7.3e-01 | -1.9e+00 | $-\infty$ | **1.2e-01** | 9.2e-02 | **1.2e-01** | 7.1e-02 |
| 361091 | 2.4e-01 | -3.0e+04 | -3.4e+00 | -3.0e+04 | **6.8e-02** | -3.1e+04 | **5.7e-02** |
| 361092 | 4.0e-02 | -2.4e+06 | $-\infty$ | **-8.7e+05** | -7.6e+09 | **-7.5e+05** | -3.7e+10 |
| 361093 | 4.3e-01 | **-3.2e-02** | $-\infty$ | -5.e-01 | $-\infty$ | **-1.1e-01** | $-\infty$ |
| 361095 | 2.2e-01 | -1.9e+01 | -1.2e+09 | **-8.6e-01** | -2.4e+07 | -1.4e+00 | **2.1e-01** |
| 361096 | 9.8e-01 | -3.7e+02 | -1.0e+08 | -3.4e+03 | **-1.8e+00** | -1.0e+03 | **4.2e-01** |
| 361098 | 8.6e-01 | -1.8e+02 | $-\infty$ | **-4.1e-01** | -1.3e+07 | -9.5e-01 | **-5.8e-01** |
| 361099 | 4.0e-01 | **-4.3e-01** | -2.3e+10 | -4.9e-01 | -2.4e+09 | **-4.8e-01** | -2.5e+08 |
| 361100 | 1.2e-01 | -9.2e+01 | $-\infty$ | -1.6e+00 | **-8.2e-02** | -3.5e+00 | **-5.2e-01** |
| 361101 | 3.4e-01 | -3.3e+03 | -1.7e+08 | **-8.1e-01** | -1.4e+02 | **-9.5e-01** | -4.7e+05 |
| 361102 | 7.5e-01 | -9.5e+02 | -1.7e+15 | **-1.3e+00** | -1.5e+15 | **-2.5e+01** | -1.4e+05 |
| 361103 | 5.4e-01 | -2.6e+00 | -1.2e+08 | 4.9e-01 | **5.1e-01** | 3.5e-01 | **4.9e-01** |
| 361104 | 6.8e-01 | **-9.e+02** | -1.0e+08 | -2.9e+03 | -3.5e+07 | **-1.4e+03** | -6.6e+05 |

Figure 5: This figure records the same information as Figure 4, but for $k = 10$.

