# OpenReview forum: "Better Private Linear Regression Through Better Private Feature Selection"
_NeurIPS.cc/2023/Conference — NeurIPS 2023 poster_

### Official Review · Reviewer_1cAJ · 2023-06-29

**Soundness:** 3 good
**Presentation:** 3 good
**Contribution:** 3 good
**Rating:** 6
**Confidence:** 4

**Summary:**

the submission proposed a private feature selection algorithm that can be directly applied as a preprocessing step before any private linear regression algorithm. The algorithm itself is based on private top-k selection with the Kendall rank correlation, which has a constant sensitivity. Further experiments on 25 regression datasets demonstrate the effectiveness of the proposed algorithm compared to a different feature selection algorithm.

**Strengths:**

1. the choice of using Kendall rank correlation as the scoring function for features is smart, and it has a constant sensitivity. With an increasing number of data samples, the algorithm becomes more and more accurate.

2. the experimental execution is thorough, and experimental results strongly supports the effectiveness of the algorithm.

**Weaknesses:**

1. the proposed private feature selection indeed improves both algorithms, but it also introduces a new hyperparameter. It would be great if the authors could provide recommendations on when to use the proposed feature selection and how to set the number of features to maintain (the submission indeed points out that 5 would be a good starting point, but I wonder if the authors could comment more).

2. it seems that the performance of BAS by itself is worse than what Tang et al reported in their work. Perhaps the authors implemented the algorithm themselves, but since BAS is based on AdaSSP, I do recommend the authors check the implementation provided by Yu-Xiang Wang here: https://github.com/yuxiangw/autodp/blob/master/tutorials/tutorial_AdaSSP_vs_noisyGD.ipynb. A few additional cares are taken in the AdaSSP's implementation.

3. Tukey itself is based on subsample and aggregate, so I think it might be interesting to integrate the feature selection step into Tukey as a whole algorithm rather than having two parts.

**Questions:**

please see the weaknesses above.

**Limitations:**

please see the weaknesses above.

---

> ### Author Rebuttal · Authors · 2023-08-09
>
> > It would be great if the authors could provide recommendations on when to use the proposed feature selection and how to set the number of features to maintain
>
> We agree that more nuanced guidance for applying DPKendall would be useful. As a first cut, a reasonable heuristic may be to use it whenever the number of features is more than 5. This suggestion is based on two considerations. First, it is uncommon for feature selection to make utility worse (see Figure 4 in the Supplement, which shows that on ⅔ of databases K-BAS is better than BAS, and on all databases K-Tukey is better than Tukey). Second, DPKendall is "cheap" in terms of both privacy and computation, as we needed only 5% of the privacy budget and (asymptotically) no more computation over OLS in most settings (see discussion at the end of Section 3.2).
>
> > It seems that the performance of BAS by itself is worse than what Tang et al reported in their work. Perhaps the authors implemented the algorithm themselves, but since BAS is based on AdaSSP, I do recommend the authors check the implementation provided by Yu-Xiang Wang … [a] few additional cares are taken in the AdaSSP's implementation.
>
> It is not obvious to us that the performance of BAS shown here is worse than that reported by Tang et al+. The only comparison present in both works is between BAS and Tukey, and both works find that BAS outperforms Tukey on a clear majority of datasets. We note that an author of the Tang et al paper examined our experiment code and pointed out that it does not add symmetric noise (as mentioned in the proof of Theorem 5.5 in Appendix A.6.3 in the first version of their paper). However, modifying our code to use symmetric noise did not noticeably affect the performance of BAS when we re-computed the median $R^2$ values, so we view this as a minor change.
>
> > Tukey itself is based on subsample and aggregate, so I think it might be interesting to integrate the feature selection step into Tukey as a whole algorithm rather than having two parts.
>
> We experimented with a similar idea while writing this paper. In the approach we tried, we run (non-private) feature selection and regression on each model, and then run the (private) Tukey mechanism on the collection of models. This enables all of the privacy budget to be spent on the Tukey mechanism. However, the downside is that the features selected differ between models, so we end up with a collection of points (models) living in different subspaces of $\mathbb{R}^d$. This worsens the performance of the Tukey mechanism, which is best when a sufficient number of models concentrate in a small volume in $\mathbb{R}^d$. Overall, we found that integrating feature selection in this way produced poor utility. Moreover, since the additional privacy cost of feature selection is small in our experiments, we suggest that the hypothetical cost of omitting this integration is small anyway.

---

> > ### Comment · Reviewer_1cAJ · 2023-08-21
> >
> > thanks for the response, and I am going to keep my score as it is.
> >
> > At this point, I only have a slightly practical concern. For datasets with high-cardinality categorical features, selecting 5 features may not be sufficient, and it may burn more privacy budget in the feature selection step. It might be informative to check the performance of the proposed feature selection step on these datasets.

---

### Official Review · Reviewer_itk8 · 2023-07-03

**Soundness:** 3 good
**Presentation:** 3 good
**Contribution:** 3 good
**Rating:** 6
**Confidence:** 3

**Summary:**

This work studies a differentially private feature selection method based on Kendall rank correlation as plug-and-play to make differentially private linear regression more suitable to high-dimensional problems without requiring users to set data bounds or algorithmic hyperparameters. This work provides e a utility guarantee for such private feature selection when features are normally distributed. This work conducts experiments across 25 datasets by applying DP-Kendall to Turkey and BAS to show the effectiveness of DP-Kendall.

**Strengths:**

1. This work makes the existing differentially private linear regression more suitable for high-dimension problems by a feature selection based on Kendall rank correlation. This work provides an analysis of how to make Kendall rank correlation DP and is also well-motivated why choosing Kendall rank correlation.

2. DP-Kendall only incurs a little additional cost to privacy and computation.

3. This work provides the utility guarantee for such DP feature selection when features are normally distributed.

4. The experiments across 25 datasets show that DP-Kendall is necessary when the low-dimension assumption does not hold.

5. The paper is well-organized and easy to follow.

**Weaknesses:**

1. DPKendall requires an end user to choose the number of features k to select. The experiment result of $k=5$ and $k=10$ shows that the value of $k$ significantly changes the results, which indicates that it may not be easy to select a proper $k$.

2. DPKendall's use of Kendall rank correlation may struggle when ties are intrinsic to the data’s structure.

Minor:
1. The experiment set-up considers $(\epsilon, \delta)=(\ln 3, 1e-5)$, while some evaluated datasets have $n>100,000$ samples, and $1e-5$ is not smaller than such $1/n$.

**Questions:**

1. The current version of the paper provides the computation cost of DP-Kendall, DP-Lasso, and standard OLS. I wonder if it is possible to compare the utility guarantee with previous works.

2. For the experiment part, it seems that the value of $k$ should not change the performance of baseline BAS and Turkey as these two do not include the feature selection step. Is the difference between these two methods in Figure 2 for $k=5$ and $k=10$ a result of random runs?


**Limitations:**

The paper states the following limitations:

1. DPKendall requires an end user to choose the number of features k to select.

2. DPKendall's use of Kendall rank correlation may struggle when ties are intrinsic to the data’s structure.

3. Kendall rank correlation may fail to distinguish between a feature with a strong linear monotonic relationship with the label and a feature with a strong nonlinear monotonic relationship with the label.

---

> ### Author Rebuttal · Authors · 2023-08-09
>
> > The current version of the paper provides the computation cost of DP-Kendall, DP-Lasso, and standard OLS. I wonder if it is possible to compare the utility guarantee with previous works.
>
> Ideally we would have a more direct comparison between DPKendall and SubLasso, but this is difficult, as the utility analysis provided in the SubLasso paper relies on several additional assumptions (see Assumption 1 and Assumption 2 in "Private Convex Empirical Risk Minimization and High-dimensional Regression", Kifer Smith Thakurta 12).
>
> > For the experiment part, it seems that the value of $k$ should not change the performance of baseline BAS and Tukey as these two do not include the feature selection step. Is the difference between these two methods in Figure 2 for $k=5$ and $k=10$ a result of random runs?
>
> This is partially due to random runs, and partially due to the $k=10$ experiment omitting some of the $k=5$ datasets for having fewer than 10 features (see the footnote in Section 4.3.1). For example, BAS obtains $R^2 > 0$ on dataset 361094, but this dataset only has 6 features and is omitted for $k=10$.

---

> > ### Comment · Reviewer_itk8 · 2023-08-13
> > **Thank you for your rebuttal**
> >
> > I have read the rebuttal, and I find it addresses most of my concerns. I would suggest in the follow-up work it would be better to set $\delta < 1/N$.

---

### Official Review · Reviewer_yU7m · 2023-07-05

**Soundness:** 4 excellent
**Presentation:** 4 excellent
**Contribution:** 3 good
**Rating:** 8
**Confidence:** 4

**Summary:**

The paper under consideration delves into the challenge of performing linear regression under differential privacy, especially in scenarios where the data is unbounded. This issue arises primarily due to the current private linear regression mechanisms that require tuning a set of hyperparameters for unbounded data, thereby degrading data privacy. To navigate this, the paper utilizes two "plug-and-play" mechanisms—namely the Turkey and the BoostedAda (BAS) mechanisms—which circumvent the necessity of parameter tuning. Nonetheless, their limitation lies in their inadequate scalability for high-dimensional data.

To address this, the paper innovates a differentially private technique for feature selection, DPKendall, intended for high dimensional unbounded data. It’s goal is to choose the most important features—those significantly correlated to the dependent variable in linear regression. These features can then be used with either the Turkey or BAS mechanism.

The distinctiveness of DPKendall lies in its utilization of Kendall rank correlation to identify correlations between features. The advantages of this method is that it does not require knowledge of data range—relying solely on the relative ordering of the data—and its bounded sensitivity, which is the measure of how much the correlation computation changes in response to a single individual's input data changing. Bounded sensitivity is crucial for upholding differential privacy.  Then, the DPKendall method privately selects k features, seeking to optimize the Kendall rank correlation with the dependent variable while minimizing its correlation with previously chosen features. The latter objective aids in eliminating redundancy.


Although principal component analysis (PCA) is a prevalent approach for dimension reduction, its applicability is restricted due to most DP methods' necessity for bounded data. Therefore, the paper compares DPKendall with another private feature selection technique, SubLasso.

The paper substantiates the efficacy of the DPKendall method through experiments on 25 datasets, each comparing different settings that apportion the privacy budget between a private feature selection method and a private linear regression method. The experimental results lend credence to the superiority of DPKendall over SubLasso in performing private linear regression, thereby highlighting its promise for this task. Moreover, the paper provides theoretical claims that the DPKendall methods is able to choose good features in the presence of many redundant features.

**Strengths:**

The paper demonstrates the applicability of the Kendall rank correlation statistical technique under differential privacy, enabling the computation of correlations within unbounded data in a privacy-preserving manner, which is of significant utility. This approach is leveraged for private feature selection, serving as a dimension reduction technique aimed at facilitating private linear regression.

The paper offers both theoretical insights and empirical evidence for their proposed method. Theoretically, it presents a novel contribution by proving that DPKendall effectively identifies features that are correlated with the dependent variable. Empirically, it substantiates the practicality of DPKendall, demonstrating that its application for feature selection culminates in superior outcomes in linear regression.


**Weaknesses:**

The experimental outcomes indicate that the DPKendall approach exhibits enhanced performance compared to the alternative SubLasso method. However, DPKendall does not invariably outperform, and the experimental specifications remain somewhat ambiguous. Consequently, it would help to discern under which conditions DPKendall delivers superior results. Addressing questions such as the impact on performance as data dimensions escalate, or which method excels at varying parameter levels could provide valuable insights.

**Questions:**

1) Could you provide a precise definition of what constitutes a "plug-and-play" algorithm? The proposed method necessitates the selection of parameter k, and also the budget split. How does this aspect impact the method's practical applicability?

2) In Section 4.2, could you elucidate how the number of gradient boosting rounds T was determined for the BAS method? Were there instances where baselines had to allocate privacy budget for determining the parameter T for the BAS method?

3) Would you clarify the execution of the K-BAS method, specifically referring to bullet 6 under Section 4.2? It is mentioned that 5% of the privacy budget was allocated to DPKendall and 95% to private linear regression. Is this accurate, or was there an allocation of 5% for model selection in this method as well?

4) Has the Kendall rank correlation been employed in previous differential privacy-related work?

5_ Are there any other potential downstream applications for private rank correlations? For instance, could it be utilized for Principal Component Analysis (PCA)?

6) How does the optimal selection of k vary depending on the dataset?

7) Could you explain the rationale behind the chosen split (5%) of the privacy budget in the conducted experiments?


**Limitations:**

yes.

---

> ### Author Rebuttal · Authors · 2023-08-09
>
> > Could you provide a precise definition of what constitutes a "plug-and-play" algorithm?
>
> See the response to Comment 1 in the general response.
>
> > The proposed method necessitates the selection of parameter k, and also the budget split. How does this aspect impact the method's practical applicability? … How does the optimal selection of k vary depending on the dataset? Could you explain the rationale behind the chosen split (5%) of the privacy budget in the conducted experiments?
>
> See the response to Comment 2 in the general response. We view the budget split as a still less important hyperparameter; the choice of 5% budget for each non-regression subroutine is meant to demonstrate that only a "small" budget is necessary for them.
>
> > In Section 4.2, could you elucidate how the number of gradient boosting rounds T was determined for the BAS method? Were there instances where baselines had to allocate privacy budget for determining the parameter T for the BAS method?
>
> We always set the number of gradient boosting rounds $T$ to 100, so this choice did not consume any privacy budget. We chose $T = 100$ based on our reading of the experiments done by Tang et al.; note that the most recent Arxiv version of their paper also uses $T = 100$ for its experiments featuring fixed hyperparameters for BAS (see their Section A.1).
>
> > Would you clarify the execution of the K-BAS method, specifically referring to bullet 6 under Section 4.2? It is mentioned that 5% of the privacy budget was allocated to DPKendall and 95% to private linear regression. Is this accurate, or was there an allocation of 5% for model selection in this method as well?
>
> We did not allocate 5% budget for model selection for K-BAS. This is because model selection only occurs when we use at least one of SubLasso or Tukey.
>
> > Has the Kendall rank correlation been employed in previous differential privacy-related work?
>
> To the best of our knowledge, the only previous DP application is by KSSW16, as mentioned in the last paragraph of the Related Work section.
>
> > Are there any other potential downstream applications for private rank correlations? For instance, could it be utilized for Principal Component Analysis (PCA)?
>
> Private rank correlation is plausibly applicable in any setting where feature selection may be useful. It is not obvious to us how it could be applied to PCA, as private rank correlation relies on the existence of discrete candidate features to evaluate against labels; in contrast, PCA (informally) identifies high-variance directions from an infinite set of candidates.

---

> ### Comment · Reviewer_yU7m · 2023-08-19
>
> Thank you for the rebuttals.
>
> I will maintain my score.

---

### Official Review · Reviewer_8Qm2 · 2023-07-08

**Soundness:** 3 good
**Presentation:** 3 good
**Contribution:** 2 fair
**Rating:** 5
**Confidence:** 4

**Summary:**

This paper studies the problem of feature selection for linear regression subject to differential privacy constraints.  One of the benefits of this work is that the feature selection does not require setting bounds on features a priori, thus allowing for fewer hyper-parameters which are difficult to set in practice.  By developing a DP version of the Kendall rank correlation, they show that first doing DP feature selection can significantly improve the quality of the DP linear regression model.  Existing DP linear regression also do not require bounds on features, so together with this DP feature selection, it provides a practical way to do DP linear regression with feature selection.

**Strengths:**

Linear regression is a fundamental problem in data analytics, so having a simple DP version of it will help practitioners in various applications.  Feature selection is an important subroutine in linear regression and reducing the hyper parameters can be beneficial.  The application of the exponential mechanism which balances the two desiderata (features that correlate with label and features that do not correlate with already selected features) is nice.  Can other works for feature selection (e.g. PCA) be summarized in a similar way?

**Weaknesses:**

This work claims that their approach will help expand the applicability of private linear regression, but it does not seem like existing approaches were too bad without feature selection.  Was a better private linear regression really the bottleneck for deploying DP linear regression in practice, especially when DP parameters are typically large in practice?  Although this is a nice result, it seems incremental.  The main contributions that I see include a sensitivity analysis of the empirical Kendall rank correlation and a utility guarantee under certain assumptions that I am not sure how the result would be used in practice as it depends on the population parameters.  I think that developing DP algorithms with as few hyper parameters is important, but the feature selection still introduces the $k$ parameter without a clear way to set it.

Further, I would like to better understand the connection between Kusner et al [22] and this work.  Is the neighboring relation between “swap” and”add-remove” technically important?  Seems like there are a few nuisances with the “add-remove” neighboring relation, e.g. the scaling of $\hat{\tau}$ and the estimation of $\tilde{n}$ in the comparison of algorithms.  I would be interested in how the various approaches in the experiments compare with as few hyper parameters as possible.  For instance, there is some budget allocated for computing $\tilde{n}$ and an $\eta$ parameter which seem arbitrary and can be avoided with the “swap” neighboring relation.

Minor:
- is the $R^2 = 0$ approach really trivial?  Don’t you need to set some bound on the labels to compute the mean?

- Line 297 “alls”

- Line 7 in Algorithm 2, should the sum be to $m$ not $n$?

- I might be misunderstanding the pseudocode in Algorithm 1, but it seems like $\hat{\tau}^X$ will always be negative.
The sensitivity of the rank correlation reminds me of work from DP non-parametric hypothesis testing from Couch, Kazan, Shi, Bray, Grace ’19.  How does the Kendal rank correlation compare with the Kruskal-Wallis rank statistic?

- In the introduction, it is not clear what is meant by “plug-and-play” requirement.  Maybe there is a missing paragraph or sentence?

**Questions:**

How important is the "add-remove" neighboring relation compared with "swap"?

Rather than using “basic” DP composition to compare the various $k$ parameters in the feature selection, can you instead frame this part under zCDP?  In the end you are quoting an approximate DP guarantee, due to the Tukey mechanism being approximate DP, so framing the feature selection as zCDP should be straightforward.  The reason is because as you increase $k$, you are suffering a larger impact on privacy loss than necessary, due to basic composition, which can also influence the general guidance you gave of using $k = 5$.  I would be interested to see the performance of the various tests using zCDP composition with $k$ being swept over several choices.

**Limitations:**

Limitations were stated in the paper.

---

> ### Author Rebuttal · Authors · 2023-08-09
>
> > The application of the exponential mechanism which balances the two desiderata (features that correlate with label and features that do not correlate with already selected features) is nice. Can other works for feature selection (e.g. PCA) be summarized in a similar way?
>
> In some sense, yes. For example, Kaprolov and Talwar's private PCA algorithm ("On differentially private low rank approximation", Kaprolov Talwar 12) uses the following process: privately estimate a rank-1 approximation, subtract the corresponding matrix from the original, and recurse on the resulting matrix. The first step is analogous to label correlation, and the second step is analogous to subtracting the influence of previous approximations. However, we note that this similarity is largely conceptual and informal.
>
> > Was a better private linear regression really the bottleneck for deploying DP linear regression in practice, especially when DP parameters are typically large in practice?
>
> See the response to Comment 1 in the general response. Briefly, we suggest that DP linear regression has, in practice, been far from a solved problem. Previous solutions can yield reasonable utility with large privacy parameters and non-private tuning, but this is still unsatisfying if we want meaningful privacy guarantees. In contrast, our methods yield good utility without large privacy parameters or non-private tuning.
>
> > Is the neighboring relation between “swap” and ”add-remove” technically important? … [f]or instance, there is some budget allocated for computing $\tilde n$ and an $\eta$ parameter which seem arbitrary and can be avoided with the “swap” neighboring relation
>
> We agree that the distinction between swap and add-remove privacy is not technically significant, and we do not suggest that the proof of Kendall rank's low sensitivity in the add-remove model is a major contribution of the paper (briefly, we frame the paper's major contributions as observing that Kendall rank correlation can be used for private feature selection, constructing a private algorithm to balance correlation and uncorrelation, proving a formal utility guarantee, and demonstrating strong empirical performance across many datasets). Nonetheless, we opted to use add-remove privacy because, for example, swap privacy allows the release of a database's size in the clear, and this can itself constitute a privacy violation in many practical settings.
>
> > Is the $R^2 = 0$ approach really trivial? Don’t you need to set some bound on the labels to compute the mean?
>
> It is trivial in the sense that it discards the features entirely. However, you are correct that privately approximating this algorithm is not trivial. We therefore suggest that the increase in $R^2 > 0$ datasets obtained via our feature selection algorithm (from 16% to 56% of all datasets) is a meaningful improvement.
>
> > Line 7 in Algorithm 2, should the sum be to $m$ not $n$? … I might be misunderstanding the pseudocode in Algorithm 1, but it seems like $\hat \tau^X$ will always be negative.
>
> Yes. Furthermore, the term $\hat Y - \frac{\hat \tau^X}{t-1}$ in Line 9 of Algorithm 1 should be a sum, not a difference. We will correct these typos in the next version.
>
> > The sensitivity of the rank correlation reminds me of work from DP non-parametric hypothesis testing from Couch, Kazan, Shi, Bray, Grace ’19. How does the Kendall rank correlation compare with the Kruskal-Wallis rank statistic?
>
> From the given paper, Kruskal-Wallis "is used to determine if several groups share the same distribution in a continuous variable". In contrast, rank correlation measures the strength of a monotonic relationship between two variables (or, in the language of Kruskal-Wallis, groups). The latter seems better suited to regression, as, for example, a feature and label typically won't have the same distribution in an OLS setup, but will have a (roughly) monotonic relationship.
>
> > In the introduction, it is not clear what is meant by “plug-and-play” requirement. Maybe there is a missing paragraph or sentence?
>
> Thanks for pointing this out. We will revise the paper to clarify. See the response to Comment 1 in the general response.
>
> > Rather than using “basic” DP composition to compare the various parameters in the feature selection, can you instead frame this part under zCDP? … as you increase $k$, you are suffering a larger impact on privacy loss than necessary, due to basic composition, which can also influence the general guidance you gave of using $k=5$.
>
> While zCDP composition technically enables better composition, this difference happens to be negligible at the parameter ranges used in our experiments. For example, we spend $\varepsilon = 0.05 \cdot \ln(3) \approx 0.06$ on feature selection. Applying basic composition with $k=10$, each call uses $\varepsilon \approx 0.006$. Applying zCDP composition and generously using half the full $\delta = 10^{-5}$ budget only results in a per-call increase in the $\varepsilon$ budget to $\varepsilon \approx 0.007$, and the change is even smaller for smaller $k$.

---

> > ### Comment · Reviewer_8Qm2 · 2023-08-19
> >
> > Thanks for addressing my comments.  I have read the rebuttal and will increase my score slightly.  I still think the results are incremental and do not itself "solve" DP linear regression. However, linear regression is a very important problem, so it is good to see progress.

---

### Author Rebuttal · Authors · 2023-08-09

We thank the reviewers for their comments. A few comments appeared in multiple reviews, so we address them in this "global" response; we would be happy to add similar clarifying language to the next version of the paper. Other specific comments are addressed in the responses to individual reviews.

> Comment 1 (Reviewer 8Qm2, yU7m): Can you elaborate on what a plug-and-play private algorithm means and why it matters in the context of regression?

The phrase plug-and-play was originally used to describe computing devices that do not require extensive user setup. We borrow this phrase to describe private algorithms with similarly low requirements for user setup. Two classic private regression algorithms help illustrate this distinction.

AdaSSP ("Revisiting differentially private linear regression: optimal and adaptive prediction & estimation in unbounded domain", Wang 18) requires a user to provide bounds on the data's feature and label norms. Users often struggle to do this, as observed in empirical user studies by SSHSV+23 ("[s]everal depositors and analysts were unsure how to fill in the lower and upper bounds of numerical variables and resorted to educated guesses that ended up being far off from the actual bounds, causing the corresponding releases to be minimally informative"; "Don't Look at the Data! How Differential Privacy Reconfigures the Practices of Data Science", Sarathy Song Haque Schlatter Vadhan 23). AdaSSP's required user bounds are therefore a meaningful setup cost that impacts utility.

A second classic private regression algorithm, DPSGD, requires a user to configure hyperparameters including learning rate, clipping norms, batch size, and number of epochs. As observed by AJRV23, "[w]hile the optimal hyperparameters [for DPSGD] consistently produce results competitive with or sometimes exceeding that of [Tukey], even the mildly suboptimal hyperparameters nearly always produce results significantly worse than those of [Tukey]" ("Easy Differential Private Linear Regression", Amin Joseph Ribero Vassilvitskii 23). This demonstrates that significant per-dataset configuration is required for DPSGD to provide good utility, even if we ignore the privacy cost of configuration.

For the reasons above, plug-and-play algorithms are important not only for ease of use but for overall utility, as careful and private tuning is often difficult for end users. This is why we suggest that this work, which broadens the applicability of existing plug-and-play private regression algorithms, is a meaningful contribution.

> Comment 2 (Reviewer yU7m, itk8, 1cAJ): What is the role of the choice of $k$?

We agree that the best possible version of our proposed algorithm would automatically choose $k$. However, we suggest that the provided experiments demonstrate that the application of DP-Kendall is relatively insensitive to the choice of $k$, in the sense that a single choice ($k=5$) yields good performance ($R^2 > 0$) on the majority of datasets. In contrast, private regression algorithms like AdaSSP and DPSGD require much more dataset-specific choices for their parameters. We note that $k=5$ and $k=10$ are the only parameters for $k$ that we evaluated, so $k=5$ is not a consequence of overfitting to the featured datasets. Finally, we also note that the comparison algorithm for plug-and-play DP feature selection, SubLasso, also requires the end user to choose $k$.

---

### Decision · Program_Chairs · 2023-09-21

**Decision:**

Accept (poster)

**Comment:**

This paper studies the problem of private linear regression. The authors propose a preprocessing DP feature selection method for two “plug-and-play" private linear regression algorithms, which is useful for data with large feature dimensions.

The reviewers are positive and agree that it is an interesting work, and easy to follow. To my understanding, this work is a good add-on to the “plug-and-play" private linear regression algorithms. However, as the reviewer point out the proposed method does not itself "solve" DP linear regression, especially for high-dimensional DP linear regression.

In light of this, I recommend acceptance and encourage the authors to incorporate the discussion about other DP linear regression methods into the revision.